# Morphological Changes in the Oral Mucous Membrane in Viral Hepatitis C Patients: A Cross-Sectional Study

**DOI:** 10.3390/ijerph19159003

**Published:** 2022-07-24

**Authors:** Vahe Azatyan, Lazar Yessayan, Aelita Sargsyan, Anna Khachatryan, Tigran Ghevondyan, Melanya Shmavonyan, Gayane Melik-Andreasyan, Kristina Porksheyan, Mikael Manrikyan

**Affiliations:** 1Department of Therapeutic Stomatology, Yerevan State Medical University (YSMU), 2 Koryun Str., Yerevan 0025, Armenia; lazyes@yandex.ru; 2Tuberculosis Research and Prevention Center, 6/2 Adonts Str., 100 Apt., Yerevan 0014, Armenia; sargsyan.aelita@gmail.com; 3Department of Pathological Anatomy, Yerevan State Medical University (YSMU), 2 Koryun Str., Yerevan 0025, Armenia; ann_khachatryan@mail.ru; 4Laboratory of Histochemistry and Electron Microscopy, After Orbeli Institute of Physiology of National Academy of Sciences of Republic Armenia (NAoS RA), 22 Brothers Orbeli Str., Yerevan 0028, Armenia; tigranghevondyan@yahoo.com; 5Department of Infectious Diseases, Yerevan State Medical University (YSMU), 2 Koryun Str., Yerevan 0025, Armenia; sh_melaniya@mail.ru; 6National Center of Disease Control and Prevention, Ministry of Health (MoH), 12 Mkhitar Heratsi Str., Yerevan 0025, Armenia; melikandreasyan@mail.ru; 7Department of Diagnostic Radiology, Yerevan State Medical University (YSMU), 2 Koryun Str., Yerevan 0025, Armenia; tina129@mail.ru; 8Department of Pediatric Dentistry and Orthodontics, Yerevan State Medical University (YSMU), 2 Koryun Str., Yerevan 0025, Armenia; dr.manrikyan@mail.ru

**Keywords:** oral mucosa, viral hepatitis C, cytokines, morphology, immunohistochemistry

## Abstract

**Background:** The objective was to reveal the most typical changes in oral mucosa in HCV patients and compare them with those in HCV negative patients. **Methods:** The study involved 96 HCV patients and 100 patients without HCV who applied to a dental clinic. The content of cytokines IL-2, IL-4, IL-10 and ɤ-INF in the oral fluid was determined by ELISA. Buccal mucosa and gums biopsies passed histological examination. An immunohistochemical study of mucous membrane biopsies was performed using monoclonal mouse antibodies to CD3+ and CD20+. **Results:** The HCV patients group included 96 (63.5% males), and the non-HCV group included 100 subjects (62.0% males) with lesions of the oral mucous membrane. The lesions of lips and oral mucosa were more frequent in HCV than in the non-HCV group—e.g., erosion (13.5% vs. 1%), cracks in the mouth corners (42.7% vs. 0%), changes in the oral mucosa surface (89.6% vs. 3.0%), hemorrhages (78.1% vs. 0%), etc. The pro-inflammatory IL-2 level was higher and anti-inflammatory IL-4 level was lower in HCV patients compared with those in the non-HCV group. **Conclusions:** Morphological changes developed in the microvasculature both worsen the tissue trophism and accelerate the healing with differentiation into coarse-fibrous connective tissue. Immunohistochemical findings indicated a decrease in local humoral immune response.

## 1. Introduction

Hepatitis C virus (HCV) is the main cause of chronic liver disease, with a global prevalence of 3% [1]. About 58 million people in the world live with chronic HCV infection, with 1.5 million newly infected people each year [1].

The extrahepatic manifestations of viral hepatitis C, caused by HCV were first spoken about in the early 1990s [2]. They were shown to develop among 74% of HCV infected individuals [3,4]. The most common HCV extrahepatic manifestations occur in the oral cavity [4]. Studies of the oral cavity in chronic liver diseases, in particular HCV, are of great interest to clinicians, since pathological processes developing in the liver usually lead to oral mucosa disorders [5]. The oral cavity can reflect liver dysfunction in many ways: gingival bleeding, increased vulnerability to bruising, petechiae, oral soreness, cheilitis and crusted perioral rash [4,6,7,8].

The individuals diagnosed with HCV are highly prone to tooth decay as well as periodontal diseases, which can affect their dietary intake, leading to a poor quality of life [6,9,10,11,12]. A meta-analysis [13] revealed increased gingival bleeding among patients with HCV compared to a non-HCV group. Gingival bleeding was shown to be a result of the edematous effect of HCV [14,15,16,17]. These findings were later confirmed by other studies exploring the relationship between HCV and oral cavity diseases [18,19,20,21]. Periodontal disorders result from a disruption in the ecological balance of the oral cavity; consequently, the periodontal bacterial pathogens induce an inflammatory reaction [17,22,23]. Other oral manifestations of HCV include black hairy tongues [23,24,25,26] and gingival desquamation [27,28].

The signs of chronic viral hepatitis in the oral cavity are inflammatory-dystrophic changes in the oral mucosa: hyperemia, dryness, edema and desquamation of the epithelium. It was shown that HCV patients exhibited generally high levels of IL-10, IL-6, IL-4 and IL-2 [29]. These cytokines are known for their immunosuppressive and anti-inflammatory changes with an effect on the oral cavity [29]. Several reviews have summarized the evidence on salivary cytokines as biomarkers for oral conditions, suggesting that the primary source of cytokines in whole saliva is gingival crevicular fluid (GCF). They have shown that several types of interleukins were present in whole saliva at higher concentrations than in major salivary gland secretions, proposing GCF to be the likely source of the cytokines [30,31]. Essentially, HCV inhibits CD4+ T lymphocyte proliferation [32,33], thus reducing the production of these important lymphocytes. This impairs the virus-specific T-cell response by altering cooperation among various components in the immune system [34,35]. Moreover, CD3 lymphocytes in HCV patients are less effective in blocking HCV replication as compared to those in healthy individuals [36].

The impaired immune system also leads to the development of oral cavity diseases among HCV patients [37,38,39]. The diagnostic methods available to the dentist include immunological [40] and morphological studies, forming a group of morphological diagnostic methods [9,11,12]. In case of a discrepancy between the preliminary clinical diagnosis and the results of histopathological examination (the gold standard of oral mucosa lesion diagnostics) [41], along with classical histological research methods [42], modern methods of immunohistochemistry should be used [43].

The relationship between HCV and oral mucosa lesions is not well established yet. The aim of this study was to reveal the most typical changes in the oral mucous membrane in patients with HCV and compare them with the oral lesions of HCV-negative patients.

## 2. Materials and Methods

### 2.1. Study Design

This study was a cross-sectional study using the medical records of patients who were hospitalized with viral hepatitis C at the “Nork” Infection hospital and Medical Clinical Centre “Armenicum” (HCV group), as well as the patients without hepatitis C (non-HCV group) who applied to the Dental Clinic N1 at Yerevan State Medical University in Yerevan, Armenia, during the January 2018 to December 2019 period.

### 2.2. General Setting

Armenia is a landlocked country at the crossroads of Southeastern Europe and Western Asia and belongs to the European Region of the WHO. The population of Armenia is about three million [44]. According to the World Bank, Armenia is an upper-middle-income country with a GDP of USD 12.6 billion and a literacy rate of 99.8% [45].

Armenia has a 3–5% prevalence of HCV among the general population, based on which Armenia is inthird place among the post-Soviet countries [46].

### 2.3. Specific Setting

The “Nork” Infection Hospital is one of the largest medical centers in Armenia, admitting approximately 6000 patients per year, providing inpatient and outpatient services for both adult and younger patients [47].

The Clinical Centre “Armenicum” provides inpatient and outpatient services to patients with viral hepatitis and human immunodeficiency virus (HIV) infection.

The Dental Clinic N1 at Yerevan State Medical University has been operating since 1995 and is the clinical base of the Faculty of Dentistry at Yerevan State Medical University. It provides therapeutic and surgical dental services, as well as orthodontic services, for both adults and children.

### 2.4. Data Collection

The study group included 96 patients with HCV, in the age range of 18–77 years, who were hospitalized in 2018 and 2019 in “Nork” Infection Hospital and Medical Clinical Centre “Armenicum” (Yerevan, Armenia). The comparison group involved 100 subjects without HCV in the age range 20–73 years, who applied to Dental Clinic N1 at Yerevan State Medical University during the same period.

The diagnosis of HCV was made via detection of viral ribonucleic acid (RNA) in the blood using polymerase chain reaction (PCR) [48]. A clinical examination to assess the condition of the oral cavity included an external examination of the lips, corners of the mouth, an assessment of various parts of the oral mucous membrane (color, surface, presence of hemorrhages and telangiectasias), as well as the condition of the tongue (color, coating and foci of epithelial desquamation) [49].

The data variables included basic demographics—age (years) and gender (male and female); clinical characteristics included—the presence of erosion on the lips (yes/no), cracks in the corners of the mouth (yes/no), changes in oral mucosa membrane surface (yes/no), hemorrhages (yes/no), telangiectasias (yes/no), coated tongue (yes/no) and foci of epithelial desquamation (yes/no)—as well as the concentrations of IL-2, IL-10, IL-4 and ɤ-INF in the oral fluid (OF).

### 2.5. Study of Cytokines in the Oral Fluid

Cytokines in the OF were tested among 45 patients with HCV and 30 patients without HCV, who agreed to pass this test. The biological material to be tested was unstimulated mixed saliva—OF, obtained without stimulation and collected with a sterile syringe into sterile Eppendorf tubes. Samples were frozen and stored at −20 °C. Then, the samples were thawed at room temperature and centrifuged at 5000 rpm in the cold. Mucin was precipitated using 6 units of Lydase per 1.0 mL of OF with our patented method (Patent RA No. 3295 A dated 16 May 2019). The concentrations of cytokines IL-2, IL-4, IL-10 and ɤ-INF were determined by the method of solid-phase enzyme-linked immunosorbent assay (ELISA) using the Vector-Best test systems (Vector-Best JSC, Novosibirsk, Russia) and were measured using a Statfax 303 Plus photometer (Awareness Technology, Inc., Palm City, FL 34990, USA).

### 2.6. Morphological Study

The materials for morphological studies were the samples of biopsy tissues excised from the mucous membrane in the area of immediate localization of the pathological process in all patients with HCV. According to the standard histological scheme, the pieces of the tissue were fixed in 10% neutral formalin, dehydrated and embedded in paraffin. A series of sections 4 µm in thickness were stained with hematoxylin–eosin and picrofuchsin by Van Gieson for a general assessment of the condition of the examined tissues. Histological micropreparations were studied with a ZEISS Primo Star trinocular microscope (ZEISS Microscopy, Jena, Germany) under 100-, 400- and 1000-times magnification. Microphotographs were taken with a ZEISS Axiocam ERc 5 s (Carl ZEISS Microscopy, Jena, Germany). All the features were examined in accordance with the international standards, WHO recommendations and recognized research methods [50].

### 2.7. Immunohistochemical Study

Immunohistochemical research was conducted with reagents produced by Zytomed (Berlin, Germany), i.e., a manual polymer detection system and a positive control. Immunohistochemical research of the oral mucous membrane biopsies was carried out using monoclonal mouse antibodies to CD3+ (clone SP7 for the determination of T-lymphocytes) and CD20+ (clone L26 for the determination of B-lymphocytes).

### 2.8. Statistical Analysis

Descriptive analyses (mean ± SD for continuous variables and frequencies/proportion for categorical variables) were computed for all variables of interest. Differences between the two groups were evaluated using “chi-square” or “Fisher’s exact” tests for categorical variables and “Wilcoxon signed rank test” for continuous variables. The Spearman correlation was performed for determination of relationships between continuous variables. *p*-value was considered significant at <0.05 and <0.001 for highly significant results. Analyses were conducted using Excel 2013 and R software.

## 3. Results

### 3.1. Clinical Examination

The HCV patients group included 96 cases: 61 males (63.5%) and 35 females (36.5%). The non-HCV group involved 100 subjects without HCV who applied to a dental clinic: 62 males (62.0%) and 38 females (38.0%). The average age in the HCV patients group was 50.1 ± 13.3, and in the non-HCV group, it was 38.0 ± 16.7. Patient complaints and data from the clinical examination of the oral cavity were taken into account when assessing the dental status, including: external examination of the lips and corners of the mouth, as well as the state of various parts of the oral mucosa (Table 1).

The objective examination of the lips in HCV patients has revealed 13 cases (13.5%) of erosion and 41 cases (42.7%) of cracks in the corners of the mouth (Table 1). There was only one case (1.0%) of erosions and cracks in the mouth corners in the non-HCV group. Changes in the oral mucosa membrane surface have been detected among 89.6% of HCV patients and in 3.0% of the non-HCV group. Some manifestations inherent in HCV were absent in the non-HCV group. Hemorrhages on the buccal mucosa and on the hard palate were observed in 75 (78.1%) HCV patients. Telangiectasias have been detected in 67.7% of patients in the HCV group. No hemorrhages and telangiectasias have been found among the patients in the non-HCV group. The examination of the tongue ofpatients in the HCV group also revealed symptoms that were absent in the non-HCV group. Coated tongue and foci of tongue surface epithelial desquamation were detected in 93.8% and 62.5% of examined HCV patients, respectively.

### 3.2. Pro-Inflammatory and Anti-Inflammatory Cytokines

Within the scope of our study, we studied the content of pro-inflammatory cytokines, IL-2 and ɤ-INF, and anti-inflammatory cytokines, IL-4 and IL-10, in the OF (Table 2).

The comparison of the concentrations of cytokines in the OF of patients in the HCV and non-HCV groups showed that the amount of pro-inflammatory cytokine IL-2 increased with a high significant difference to 26.0 ± 17.9 (9.2 times, *p* < 0.001). The level of anti-inflammatory cytokine IL-4 decreased with the same significant difference (*p* < 0.001) of 0.2 ± 0.8 (71.5 times). The amount of IL-10 also significantly increased to 3.6 ± 6.6 (3.9 times, *p* = 0.0267). The increase in ɤ-INF in HCV patients was statistically insignificant in comparison with the non-HCV group (*p* = 0.113) (Figure 1).

### 3.3. Pathohistological and Immunohistochemical Study

Pathological processes of the oral cavity were mainly localized on the buccal mucous membrane, the tongue and lips. Five groups of major pathomorphological changes were identified in the mucous membrane in HCV patients, such as: inflammatory infiltration (lymphoplasmocytic or plasmocytic infiltration with admixture of neutrophils), circulatory disorders, mucosal ulceration with fibrinous film, mucosal fibrosis and dystrophic changes in the squamous epithelium. The inflammatory reaction was observed in all morphologically examined patients with HCV and detected in the form of lymphoplasmocytic infiltration in 89.6%. The inflammation was predominantly of the proliferative type and was localized mostly in the upper parts of the mucous membrane. Lymphoid infiltration was also found around unevenly congested micro-blood vessels at the epithelium border with the underlying tissue (Figure 2a) and in some cases with the migration of inflammatory infiltrate cells into the thickness of the epithelial layer (Figure 2b).

Circulatory disorders were revealed in 100% of the examined patients in the form of edema, hemorrhages of various sizes due to destruction of the blood vessel walls, stasis in capillaries, marginal standing of blood cells corpuscles in venules and capillaries, hyperemia and angiomatosis (Figure 2c). Obliteration of the vascular lumina, fibrinoid necrosis and fibrinoid swelling of the vessel walls were observed. Hyperplastic, metaplastic and dystrophic changes in the squamous epithelium in the form of acanthosis, parakeratosis and thickening, in comparison with normal mucosa (Figure 2d), were revealed in 100% of the examined HCV patients.

Fragments of necrotic bone tissue, most likely due to sequestration of the jawbone, were found in few patients (6.3%) with HCV. Damage of the epithelial cells was between cytoplasmic vacuoles up to balloon dystrophy, death and desquamation of the epithelium with formation of micro-erosions. Comparing these five pathomorphological changes showed that mucous membrane fibrosis was detected in 96% of HCV patients with a significant difference from the non-HCV group (*p* < 0.001). Changes caused by the development of mucous membrane sclerosis were determined in 100% of the cases in the HCV group.

Immunohistochemical research of the biopsies of mucous oral membrane taken from patients with HCV led us to evaluate the quantitative composition of infiltrate to T-lymphocytes (CD3+) and B-lymphocytes (CD20+).

Diffuse lymphocytes in the plate mucous oral membrane are represented mainly by T-cells, though T-lymphocytes were singly localized in the thick epithelial stratum. B-lymphocytes were diffuse in scanty quantity. Single plasmocytes were also scattered in the infiltrate, mainly in the surface part of the mucous membrane under the epithelium (Figure 3a,b).

Thus, in the patients with HCV, we saw single diffuse CD20+ lymphocytes, which is evidence of a local weakly expressed humoral immune response.

## 4. Discussion

A comprehensive clinical, morphological, immunohistochemical and immunological study of the oral mucosa in the HCV and non-HCV patients was performed.

According to the results of our studies, the following lesions of the oral mucosa were detected: cracks in the corners of the mouth in 42.7% of cases, changes in the oral mucosa membrane surface—89.6%; hemorrhages on the buccal mucosa and hard palate—78.1%; telangiectasias on the buccal mucosa—67.7%; coated tongue—93.8%; and desquamation of the epithelium of the tongue—62.5%. Cozzani E. et al. (2021) conducted a dental examination of patients with chronic diffuse liver diseases, including viral etiology (HCV), which showed that damages of the oral mucosa were observed in 75% of patients. The structure of diseases of the oral mucosa in these patients was represented by various manifestations, such as stomatitis of various localization, red lichen planus and candidiasis of the oral cavity [38]. In a study by Helenius-Hietala J. (2014), in patients with liver diseases who should undergo liver transplantation, lesions of the oral mucosa in the form of candidiasis, herpes infections of the first and second types, folded tongue and oral cavity cancer were observed [51]. In the available literature, we practically did not find studies which used different criteria to simultaneously assess the state of various sections of the oral mucosa and the tongue.

We have tested the levels of cytokines in the OF and found that in the group of patients with HCV, compared with the non-HCV group, the level of pro-inflammatory cytokineIL-2 and anti-inflammatory IL-10 increased and the level of anti-inflammatory cytokine IL-4 decreased. The increase in the level of ɤ-INF in the oral fluid in HCV patients compared with the non-HCV group was insignificant. These findings were consistent with the existing literature; e.g., in their research, Ribeiro C.R.A. et al. (2021) found high serum levels of IL-10, IL-4, IL-2 and TNF-α in HCV patients compared with controls [11].

The results of our research have shown circulatory disorders and inflammation in the oral mucosa, which were revealed in 100% of the examined patients. In cases of HCV, significant morphological changes developed in microvascular channels had dual influence. On the one hand, impaired blood circulation affects the tissue trophism; on the other hand, the high density of blood vessels in the regenerating granulation tissue ensured acceleration of metabolic processes, which promoted healing and differentiation into coarse-fibrous connective tissue. Azatyan V. et al. (2021) performed a morphological study of oral mucosal biopsies in patients with HBV, HCV and HIV that also revealed general morphological changes in the oral mucosa [52].

An immunohistochemical study revealed a significant decrease in CD3+ and CD20+ lymphocytes in patients with HCV which indicates a decrease in local immune responses. Available studies have been dedicated to the study of minor salivary glands in patients with HCV suffering from Sjögren’s syndrome with the use of monoclonal antibodies to CD3, CD8, CD20 and HLA-DR [53,54].

For the future, deeper immunological studies related to oral mucosa lesions in viral hepatitis are planned.

Our study has strengths and limitations. One of the strengths of the research is that, to the best of our knowledge, there was no similar research performed in Armenia before.

In addition, the study used a comprehensive approach looking at the lesions of oral mucosa from clinical, biochemical, pathohistological and immunohistochemical points of view. Thus, the paper contains more detailed results which are few or lacking in the similar publications available; i.e., our research is the first to conduct an immunohistochemical study of biopsies of the oral mucosa in patients with HCV to assess the qualitative composition of the infiltrate to T-lymphocytes (CD3+) and B-lymphocytes (CD20+).

One of the limitations of the study was that, despite the fact that the HCV group had 96 participants and the non-HCV group had 100, only 45 patients from the HCV group and 30 patients from the non-HCV group agreed to undergo the test for cytokines in the oral fluid (OF).

Another limitation was that the initial raw data (from which the Excel database was created) were from a paper-based registry; as the data were not double entered in the process of data transmission, errors could have happened.

## 5. Conclusions

Our study has shown that HCV contributes to the injury of the oral mucous membrane. Everybody in the HCV group developed circulatory disorders and inflammation in the oral mucosa. The immunohistochemical study revealed a decrease in CD20+ lymphocytes in the biopsies of patients with HCV, which also indicate the decrease in local humoral immune responses. Further research is necessary to better study the levels of ILs in HCV patients and their associations with oral mucosa lesions.

## 6. Patents

During the study, we obtained a patent for discovering the method of precipitation of mucin from oral fluid using 6 units of Lydase per 1 mL of oral fluid before centrifugation in the cold (Patent RA No. 3295 A dated 16 May 2019, authors: Lazar Yessayan and Vahe Azatyan).

## Figures and Tables

**Figure 1 ijerph-19-09003-f001:**
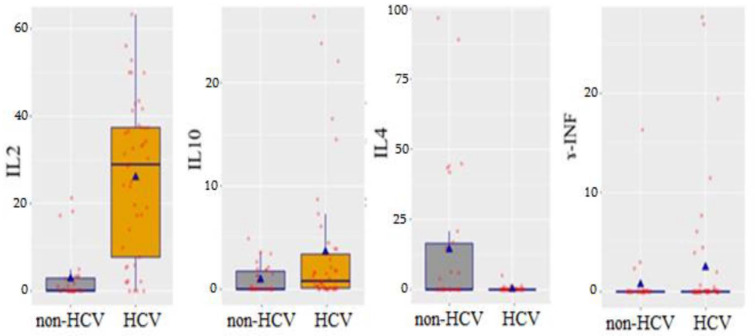
Oral fluid cytokine levels (IL-2, IL-10, IL-4 and ɤ-INF) in HCV patients and non-HCV group.

**Figure 2 ijerph-19-09003-f002:**
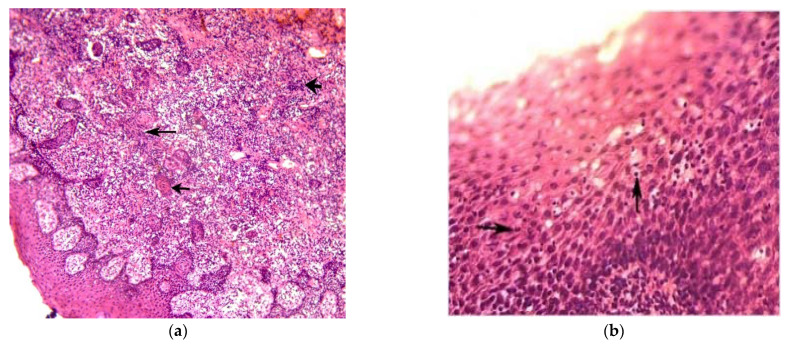
(**a**) Lymphoplasmocytic infiltration of the lamina propria of the oral mucosa in the patient with HCV. The large bold arrow marks the site of intense lymphoplasmocytic infiltration. Cell identification was carried out at higher microscope magnifications. A thin, long arrow marks the site of incipient sclerosis of the lamina propria. A thin, short arrow indicates a dystrophically altered apex of the acanthotic strand of the epithelial cover (stained with hematoxylin–eosin, ×100). (**b**) Migration of the cells of the inflammatory infiltrate into the thickness of the epithelial layer of the oral mucosa in the patient with HCV (arrow; stained with hematoxylin–eosin, ×400). (**c**) A section of the own plate of the oral mucosa in a patient with HCV. There is swelling of the connective tissue with hemorrhages in the thickness of the tissue (thick, short arrow), dilatation and a plethora of capillaries (short and long, thin arrows). Around the longitudinally cut, full-blooded blood vessel, the presence of mononuclear inflammatory cell infiltration is shown (thin, long arrow) (stained with hematoxylin–eosin, ×400). (**d**). A section of the oral mucosa in the patient with non-HCV. This material served as a control. The bold arrow indicates one of the two intact minor salivary glands with lobular ducts. The long, thin arrow indicates intact stratified squamous nonkeratinized epithelium. The short, thin arrow points to an intact lamina propria composed of delicate bundles of collagen fibers. There is scant to moderate mononuclear inflammatory cell infiltration in the left below part of lamina propria (stained with hematoxylin–eosin, ×100).

**Figure 3 ijerph-19-09003-f003:**
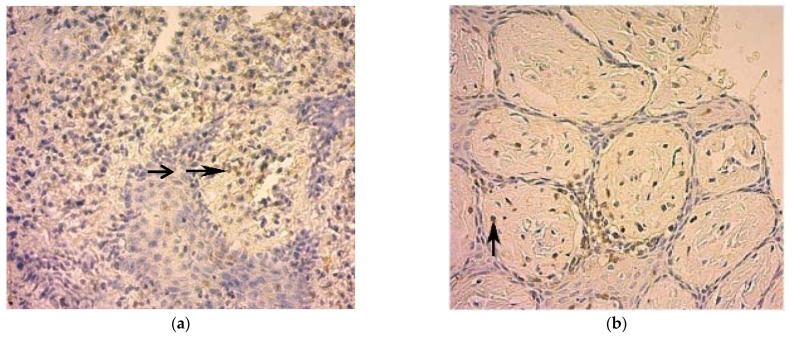
(**a**,**b**). Diffuse positive reaction to CD3+ in the cells of the inflammatory infiltrate (horizontal arrow) in the oral mucosa in patients with HCV (×100) (**a**). Focal positive reaction to CD20+ in scattered single cells of the inflammatory infiltrate (vertical arrow) of the oral mucosa in the patient with HCV (×400) (**b**).

**Table 1 ijerph-19-09003-t001:** Clinical examination data of the oral mucosa in patients with HCV and in the non-HCV groups.

Sign	Non-HCV *n* = 100		HCV *n* = 96		*p*-Value *
	Absolute Number	%	Absolute Number	%	
Erosion on the lips					<0.001
No	99	99	83	86.5	
Yes	1	1	13	13.5	
Cracks in the corners of the mouth					<0.001
No	99	99	55	57.3	
Yes	1	1	41	42.7	
Changes in the oral mucosa membrane surface					<0.001
No	97	97	10	10.4	
Yes	3	3	86	89.6	
Hemorrhages on the buccal mucosa and the hard palate					<0.001
No	100	100	21	21.9	
Yes	0	0	75	78.1	
Telangiectasias on the buccal mucosa					<0.001
No	100	100	31	32.3	
Yes	0	0	65	67.7	
Coated tongue					<0.001
No	100	100	6	6.2	
Yes	0	0	90	93.8	
Foci of epithelial desquamation on the surface of the tongue					<0.001
No	100	100	36	37.5	
Yes	0	0	60	62.5	

* *p*-value test result from the comparison between non-HCV and HCV groups.

**Table 2 ijerph-19-09003-t002:** Oral fluid cytokine levels in the non-HCV group and in patients with HCV (mean ± SD).

Cytokines	Non-HCV (*n* = 30)	HCV (*n* = 45)	Odds Ratio	95% CI	*p* Value *
IL2	2.83 ± 5.67	25.99 ± 17.86	−23.17	[−28.89; −17.44]	<0.001
IL10	0.94 ± 1.33	3.63 ± 6.58	−2.69	[−4.72; −0.66]	0.0267
IL4	14.29 ± 26.11	0.2 ± 0.79	14.09	[4.34; 23.84]	<0.001
ɤ-INF	0.72 ± 3.04	2.46 ± 6.52	−1.74	[−3.98; 0.49]	0.113

* *p*-value test result from the comparison between non-HCV and HCV groups.

## Data Availability

Not available.

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
