# Peer review of "Morphological Changes in the Oral Mucous Membrane in Viral Hepatitis C Patients: A Cross-Sectional Study"

_ijerph, 2022, doi:10.3390/ijerph19159003_

Round 1
Reviewer 1 Report
The manuscript of “Morphological changes in the oral mucous membrane in Viral Hepatitis C patients, a cross-sectional study” is reviewed that the relationship between symptoms of oral mucosal disease and blood cytokine levels in HCV-positive patients. Oral symptoms in HCV-positive patients have been reported before and described in paper or clinical report. In this manuscript, the author reported the symptom of mucosa and also cytokines, and found significant differences increasing IL-2 and IL-10 and decreased CD4+ lymphocytes. Although the paper is of interest, several points are noted and are presented below.
Materials and methods
P3L104: How the author deciding the oral findings of the list “data collection” in every patient? Is there a reference cited of the method evaluation standard?
P3L104: Did the author recognize the clinical findings (positive count) in same place every patient? Or did you consider it positive count if there was even one spot on the mucosa anywhere (buccal, palate, gingiva, tongue mucosa)?
P4L144: Please describe the method what the author referenced “WHO recommendations and recognized research methods”
Table1:The described “Coated tongue”, What is the patient's degree of plaque control? The patients were intervened oral care by professional cleanings before collect samples?
Table1: How decide criterion for “Changes of oral mucosa membrane surface”?
Figure1: It is difficult to find the detail deference. The resolution is better to clearly.
Figure2: the picture of “a” was observed an artifact and crack. The cell infiltration was observed but not observed capillary dilation. It is better to change clearer picture.
P4L205: The author describes “lymphoplasmacytic infection”, the cells were identified?
Discussions
How the author recognized the relationship between HCV positive and mucosal lesion, which like lichen planus?
The data was noted that CD4 positive cells was decreased, is it possible to display to numerical value? For instance, the difference positive cell number in each field of microscopic vison.
How HCV positive patients were decisioned. The value of HCV amount was measured?
Reviewer 2 Report
The authors investigated typical morphological changes in the oral mucosa of HCV-positive and HCV-negative patients. In addition, they performed histological and immunohistochemical examinations of the oral mucosa and quantified several cytokines in saliva. These results seem to be an interesting study showing specific changes in the oral mucosa in HCV-infected patients. Since this is a cross-sectional study, it may not be possible to mention the causes and mechanisms that develop the lesions, but several items should be further considered.
1. The authors examine CD3 and CD20 markers for infiltrated cells in the oral mucosa, but it would be better to check CD4, CD8, CD56, CD68 and other markers as well to obtain more information about inflammation.
2. Figures 2 and 3 show HCV-positive patients, but they are qualitative and unclear whether there are many infiltrating cells, it would be better to add an HCV-negative figure as a control.
3. What was the amount of HCV virus in the saliva from HCV-positive patients?
4. What were the HCV viral RNA molecules in the oral mucosa in HCV-positive patients?
5. Can you please discuss where the cytokines observed in the saliva were originated from?
6. The cytokine levels in saliva have been tested, but what about the cytokine levels in the blood?
